# *leafkin*—An R package for automated kinematic data analysis of monocot leaves

Jonas Bertels[1] and Gerrit T.S. Beemster[1] (iD)

[1]Integrated Molecular Plant Physiology Research Group (IMPRES), Department of Biology, University of Antwerp, Antwerp, Belgium

## Original Research Article

cell division; cell elongation; elongation zone; growth zone; kinematic analysis; meristem; monocotyledonous leaf; R.

**Author for correspondence:** Gerrit T. S. Beemster,
E-mail: gerrit.beemster@uantwerpen.be

### Abstract

Growth is one of the most studied plant responses. At the cellular level, plant growth is driven by cell division and cell expansion. A means to quantify these two cellular processes is through kinematic analysis, a methodology that has been developed and perfected over the past decades, with in-depth descriptions of the methodology available. Unfortunately, after performing the lab work, researchers are required to perform time-consuming, repetitive and error-prone calculations. To lower the barrier towards this final step in the analysis and to aid researchers currently applying this technique, we have created *leafkin*, an R-package to perform all the calculations involved in the kinematic analysis of monocot leaves using only four functions. These functions support leaf elongation rate calculations, fitting of cell length profiles, extraction of fitted cell lengths and execution of kinematic equations. With the *leafkin* package, kinematic analysis of monocot leaves becomes more accessible than before.

## 1. Introduction

The effect of genetic modification and the impact of biotic or abiotic stress on plants is frequently evaluated by measuring growth. Growth is often quantified on whole plant (e.g. dry mass) or organ (e.g. root or leaf length) level (Erickson, 1976; Poorter & Garnier, 1996). However, it represents the combined result of two processes at the cellular level, that is cell division and cell expansion (Beemster et al., 2003). Therefore, various studies have quantified these cellular processes, often linking them to data from biochemical and molecular assays for a more mechanistic understanding of different growth responses (Sprangers et al., 2016). The importance of growth analysis at the cellular level is clearly demonstrated by a meta-study by Gázquez and Beemster (2017), who identified the regulation of the transition from cell division to cell expansion as the key cellular mechanism for organ size regulation.

Monocotyledonous leaves are ideally suited for the quantification of cell division and expansion, because they are linear, steady-state growing organs. This means that, for a certain period during their development, a growth zone with a stable meristem and elongation (expansion in longitudinal direction) zone size is present at the base of the leaf, resulting in an approximately constant leaf elongation rate (Muller et al., 2001; Schnyder et al., 1990). We consider the maize leaf an ideal model organ to study leaf growth regulation because it hosts a large growth zone, providing ample material for biochemical and molecular analyses in relation to cellular growth responses (Avramova et al., 2015).

The methods of plant growth analysis have made considerable progress over the past century. In the classical approach, which started to evolve in the 1920's (Blackman, 1919; West et al., 1920), the relative growth rate is calculated by dividing the difference in ln-transformed plant weight over time (Poorter & Garnier, 1996). Two decades later, Sinnot (1939) pointed out that transparent root meristems could be studied under water immersion lenses, where drawings made at intervals from one to several hours allowed researchers to track cell division by the formation of new cell walls and cell elongation by changes in cell sizes. Sinnott's publication was followed by the work of Goodwin & Stepka (1945) and Erickson & Sax (1956), who developed a more mathematical foundation for the determination of cell division and cell elongation rates by combining velocity fields and cell length profiles in roots. Later, in the late 1970s and early 1980s, the foundation for kinematic growth analysis was laid by applying equations from fluid dynamics to describe plant organs as linear structures with a flux of cells (or substances such as minerals) passing at each position determined by local velocity and density (Gandar, 1980; Silk, 1984; Silk & Erickson, 1979). Growth zones are composed of meristem, a region of small diving cells, and the elongation zone, where cells rapidly increase in cell size due to cell expansion. Cells are displaced by cell division and cell elongation until they stop growing and enter the mature part of the leaf.

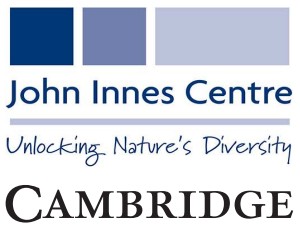

The kinematic analysis for the study of organ growth has been adopted by a limited number of laboratories [summarized by Gázquez and Beemster (2017) and Sprangers et al. (2016)]. In the past decade, a considerable effort was undertaken to make the methodology more accessible for non-specialized labs by detailed method descriptions for the kinematic analysis of roots and leaves (Nelissen et al., 2013; Rymen et al., 2010). More recently, a video tutorial was published, demonstrating step by step how to perform a kinematic analysis on maize and other monocot leaves (Sprangers et al., 2016). A significant difficulty that remains when performing this kinematic analysis on monocot leaves, is the processing of raw data and a correct application of mathematical equations involved. In our experience, the analysis of the acquired data can be daunting and the repetitive manual processing of a large number of measurements is error-prone.

To help novices with the application of a kinematic analysis on monocotyledonous leaves and to simplify and accelerate the work of researchers already employing this technique, we developed *leafkin*, a simple to use R-package, which performs all required calculations using only four functions. Once familiar with these four functions, time required for kinematic data analysis is reduced to a couple of minutes and human errors in the analysis are avoided (e.g. selecting wrong cells in Excel), while errors in the input data are more easily identified (e.g. by inspecting cell length plots). Also, a user manual is provided as supplementary material (Supplementary File S1), which is accompanied by a full example dataset and tutorial script (available on https://github.com/impres-lab). These can be used to familiarize new users with the required datasets and *leafkin* functions, prior to analysing their own datasets.

In Section 2, we describe the required datasets and the used methodology for each of the functions. In Section 3, we illustrate the use and outcome of the functions with special attention to parameter settings for more control on the generated output. In Section 4, we highlight the advantages and limitations of the package.

## 2. Methods

### 2.1. User manual

In the user manual (Supplementary File S1), the practical steps of the kinematic analysis are introduced in more detail. It also provides a flowchart that illustrates the links between the collected data and *leafkin* functions. Hereafter, it provides more details on the requirements for, and installation of *leafkin*, followed by a step-by-step description of its use in the kinematic data analysis. Next, the manual discusses potential errors and provides additional information on tidy and wide data (data formats which are referred to in this article). Finally, all formulas used by the *leafkin* library are presented in the manual.

### 2.2. Practical steps of kinematic analysis

In short, in order to study the growth of a specific leaf at the cellular level by kinematic analysis, around 15 plants are required for each treatment/genotype to be studied. First, the length of the leaf is measured, starting after it emerges from the whorl of older leaves, usually on a daily basis. After tracking leaf growth for a couple of days (in maize: at least 3 days), 5–7 plants are dissected during the period of steady-state growth for microscopy, allowing the remainder of the plants to reach their final leaf length. During the dissection, the growth zone of the leaf of interest is isolated

(e.g. the basal 10 cm of a maize leaf, that is starting where the leaves are attached to the stem). In this growth zone, meristem size [through 4′,6-diamidino-2-phenylindole (DAPI) staining of the nuclei] and cell length profiles are determined (Sprangers et al., 2016; Supplementary File S1).

### 2.3. Required datasets

The practical work results in three datasets, that is leaf lengths, cell lengths and meristem sizes, all of which are required by *leafkin*. The raw data can be entered in a spread sheet program (e.g. Microsoft Excel), but needs to be saved as tab-delimited text files. We advise to use this format because importing Excel files directly into R may transform date–times into numbers, rendering them unusable by the *leafkin* functions.

The leaf length data file requires a column with unique plant IDs, followed by multiple columns containing leaf length measurements, expressed in millimetres (Table 1a). The first row contains the headers, which should be *plant_id* for the first column, while the following column headers are in the date-time format *yyyy/mm/dd hh:mm* (or *yyyy/mm/dd hh:mm:ss*), indicating when measurements were made.

The cell length measurements should be organized in three columns (Table 1B). The first column (header = *plant_id*), holds the plant ID for each measurement. The second column (header = *position*) contains the position of the cell length measurements relative to the leaf base (in centimetres) and is followed by the cell lengths themselves (in micrometres) in the third column (header = *cell_length*). Cell length measurements of all plants are combined in these three columns.

The third file should contain the meristem size measurements (Table 1c). The first column (header = *plant_id*), contains the unique plant IDs, whereas the second column (header = *mer_length_um*), contains meristem sizes (in micrometres).

It is important to note that units and column names should be strictly respected. Also, plant IDs should be identical across all three files, since these are used to combine the data originating from the different measurements.

### 2.4. Software requirements

The *leafkin* package works with R version 4.0.0 or higher (R Core Team, 2014). Windows users are advised to install Rtools40 (a toolchain bundle which aids building R packages locally) in order to install *leafkin* without warnings related to Rtools (https://cran.r-project.org/bin/windows/Rtools/). Installing RStudio, an integrated development environment for R, is recommended to increase the ease of use of R code (a free open source edition is available on https://rstudio.com/; RStudio Team, 2015).

### 2.5. Sample data, tutorial script and **leafkin** *installation*

A sample dataset and tutorial R-script are available on the IMPRES-lab GitHub page (https://github.com/impres-lab). Sample data originated from a kinematic analysis, performed in Bertels et al. (in press). We highly recommend first time users to download the sample data and tutorial script and use these in conjunction with the user manual (Supplementary File S1).

The *leafkin* package is maintained on the IMPRES-lab GitHub page. Prior to *leafkin* installation, the *install_github()* function from the *devtools* package is used to install *leafkin* directly from

**Table 1.** Example data and column description for the datasets required for kinematic analysis using *leafkin*.

### A. Leaf length measurements data and column descriptions

Example data

| plant_id | 2016/12/13 10:00 | 2016/12/14 10:00 | 2016/12/15 10:00 | 2016/12/16 10:16 | 2016/12/17 10:00 |
|----------|------------------|------------------|------------------|------------------|------------------|
| C.1 | 142 | 216 | 293 | | |
| C.2 | | 142 | 212 | 296 | |
| C.3 | | 196 | 277 | 360 | 436 |
| C.4 | | 194 | 268 | 352 | |

**COLUMN DESCRIPTION**

| COL | Header | Type | Brief description |
|-----|--------|------|-------------------|
| 1 | plant_id | char or int | Contains the plant ID for which leaf lengths were measured. |
| 2-LAST | data time format yyyy/mm/dd hh:mm(:ss) | int or double | Contains leaf length measurements in millimetre on a certain day-time. Time can be in hh:mm or hh:mm:ss. |

### B. Cell length measurements data and column descriptions

Example data

| plant_id | position | cell_length |
|----------|----------|-------------|
| C.1 | 0.01 | 27.18 |
| C.1 | 0.01 | 23.71 |
| C.1 | 0.01 | 23.68 |
| C.1 | 0.01 | 22.23 |

**COLUMN DESCRIPTION**

| COL | Header | Type | Brief description |
|-----|--------|------|-------------------|
| 1 | plant_ID | char or int | Contains the plant ID for which leaf lengths were measured. |
| 2 | position | int or double | Contains the position at which leaf lengths were measured in centimetre. |
| 3 | cell_length | int or double | Contains cell length measurement in micrometre. |

### C. Meristem length measurements data and column descriptions

Example data

| plant_id | mer_length_um |
|----------|---------------|
| C.1 | 12423 |
| C.2 | 14792 |
| C.4 | 12350 |
| C.7 | 14568 |

**COLUMN DESCRIPTION**

| COL | Header | Type | Brief description |
|-----|--------|------|-------------------|
| 1 | plant_ID | char or int | Contains the plant ID for which leaf lengths were measured. |
| 2 | mer_length_um | int or double | Contains the length of the meristem in micrometre. |

A: Example of leaf length measurements data and column descriptions. B: Example of cell length measurements data and column descriptions. C: Example of meristem length measurements data and column descriptions. The types char, int and double refer to respectively characters (i.e. everything which includes letters, or numbers specified to be handled as letters), integers (i.e. numbers without decimals) and double (i.e. numbers which can contain decimals).

the GitHub repository (i.e. *devtools::install_github("impres-lab/leafkin")*, more details in the user manual, Supplementary File S1).

## 2.6. leafkin *user functions*

The user functions of the package are *calculate_LER()*, *get_pdf_with_cell_length_fit_plots()*, *get_all_fitted_cell_lengths()* and *kinematic_analysis()*. These four functions allow the user to perform all calculations needed to perform a kinematic analysis of monocotyledonous leaves.

### 2.6.1. calculate_LER(). *calculate_LER()* calculates the leaf elongation rate (LER) for each plant using the leaf length measurements

(formula 1 in the user manual, Supplementary File S1) and will, by default, output the mean values for each plant using the first two time-intervals. The user can specify three parameters, that is *leaf_length_data*, *n_LER_for_mean* and *output*. *leaf_length_data* is the parameter to which the imported leaf length data have to be assigned. These leaf length data must be imported into R beforehand as a data.frame or tibble (a modern format of a data.frame). Next, *n_LER_for_mean* indicates how many intervals with corresponding LERs are to be used to calculate the mean LER (default = 2), starting from the first measurement. In case a number larger than the number of LERs available is specified, only the available intervals will be used (Table 2a,b). Finally, *output* determines the format of the output of the function. By default,

**Table 2.** The output of the *calculate_LER()* function.

**A**

| plant_id | mean_plant_LER [mm/h] |
|---|---|
| C.1 | 3.145833 |
| C.10 | 3.1875 |
| C.11 | 3.145833 |
| C.2 | 3.208333 |
| C.3 | 3.416667 |
| C.4 | 3.291667 |
| C.5 | 3.125 |
| C.6 | 3.208333 |

**B**

| plant_id | mean_plant_LER [mm/h] |
|---|---|
| C.1 | 3.145833 |
| C.10 | 3.1875 |
| C.11 | 3.239583 |
| C.2 | 3.208333 |
| C.3 | 3.364583 |
| C.4 | 3.291667 |
| C.5 | 3.09375 |
| C.6 | 3.28125 |

**C**

| plant_id | date_and_hour | leaf_length [mm] | time_hours [h] | growth_mm [mm] | LER [mm/h] |
|---|---|---|---|---|---|
| C.1 | 2016/12/13 10:00 | 142 | | | |
| C.1 | 2016/12/14 10:00 | 216 | 24 | 74 | 3.083333333 |
| C.1 | 2016/12/15 10:00 | 293 | 24 | 77 | 3.208333333 |
| C.10 | 2016/12/14 10:00 | 157 | | | |
| C.10 | 2016/12/15 10:00 | 229 | 24 | 72 | 3 |
| C.10 | 2016/12/16 10:00 | 310 | 24 | 81 | 3.375 |
| C.11 | 2016/12/14 10:00 | 151 | | | |
| C.11 | 2016/12/15 10:00 | 221 | 24 | 70 | 2.916666667 |
| C.11 | 2016/12/16 10:00 | 302 | 24 | 81 | 3.375 |
| C.11 | 2016/12/17 10:00 | 382 | 24 | 80 | 3.333333333 |
| C.11 | 2016/12/18 10:00 | 462 | 24 | 80 | 3.333333333 |

**D**

| plant_id | date_and_hour | leaf_length [mm] | time_hours [h] | growth_mm [mm] | LER [mm/h] |
|---|---|---|---|---|---|
| C.1 | 2016/12/13 10:00 | 142 | | | |
| C.1 | 2016/12/14 09:23 | 216 | 23.38333333 | 74 | 3.164647185 |
| C.1 | 2016/12/15 08:16 | 293 | 22.88333333 | 77 | 3.364894392 |
| C.10 | 2016/12/14 09:23 | 157 | | | |
| C.10 | 2016/12/15 08:16 | | 22.88333333 | | |
| C.10 | 2016/12/16 10:00 | 310 | 25.73333333 | 153 | 3.147068906 |
| C.11 | 2016/12/14 09:23 | 151 | | | |
| C.11 | 2016/12/15 08:16 | 221 | 22.88333333 | 70 | 3.058994902 |
| C.11 | 2016/12/16 10:00 | | 25.73333333 | | |
| C.11 | 2016/12/17 10:00 | | 24 | | |
| C.11 | 2016/12/18 10:00 | 462 | 24 | 241 | 3.268535262 |

**A.** The output of the *calculate_LER()* function with the number of LERs (*n_LER_for_mean*) set to 2, that is all plants have enough measurements to support the calculation of the LER mean. **B.** The output of the *calculate_LER()* function with the number of LERs (*n_LER_for_mean*) set to 4, that is plants harvested for microscopy analysis only have two LERs, though the function still correctly calculates the mean LER, while for plants with more measurements, the LER changes (dotted red arrows) because more calculated LERs are incorporated in the mean value. **C.** The output of *calculate_LER()* with the *output* parameter set to *tidy_LER*, allowing access to the individually calculated LERs. **D.** Illustration of how the *calculate_LER()* function handles variable time intervals (i.e. not all 24 h time intervals) and missing data. Notice how time intervals, growth intervals and LERs are corrected accordingly (full red arrows).

*output* is set to '*means*', causing the *calculate_LER()* function to return mean LER for each plant. However, the user can also choose to set the *output* parameter to '*tidy_LER*' and '*wide_LER*', which will result in returning a tibble containing all calculated LERs, either in a tidy (Table 2c) or wide format, respectively (see user manual, Supplementary File S1, for more information on tidy and wide data formats). These can be used to visualize the LER over time (useful to check the steady-state assumption during the period used to calculate the average LER).

Typically, leaf lengths are measured once a day, however, using multiple measurements per day is also possible (note: in this case, consider increasing *n_LER_for_mean* to cover a sufficiently large time-interval). During LER calculation, the function skips time-points with missing measurements and adjusts the corresponding

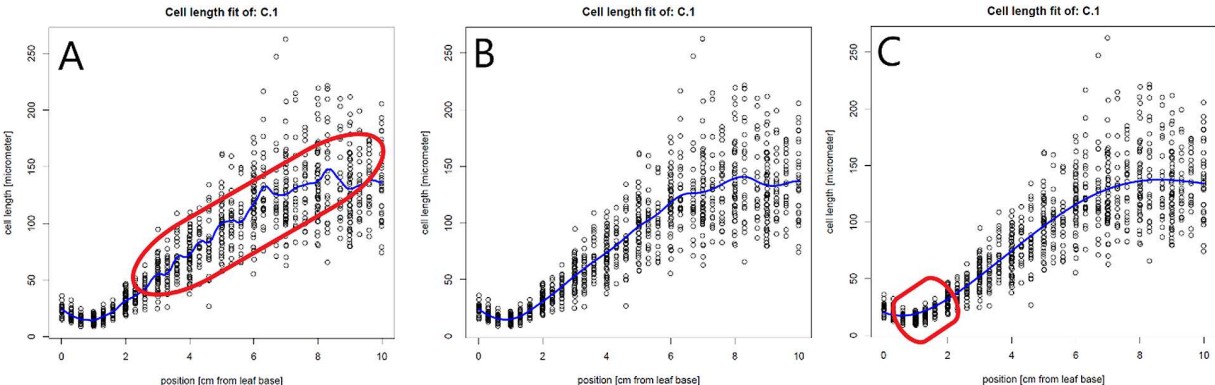

**Fig. 1** The effect of the bandwidth multiplier parameter on cell length fits. (a) A very strict fit of the cell lengths by setting the *bw_multiplyer* to 0.3. A strict fit can result in too much variation in the fit (encircled in red). (b) Fitted cell length data, using the calculated bandwidth (bandwidth multiplier = 1).(c) A more loose fit of the cell lengths by setting the *bw_multiplyer* to 3. A loose fit can result in oversmoothing and thereby poor fitting of the cell sizes, especially at the end of the meristem (encircled in red) and/or the end of the growth zone.

time-intervals accordingly, ensuring that the function can handle missing data (Table 2d). The calculated LERs and mean LERs are stored within the function and depending on how the user specified the *output* parameter, mean LERs or all LERs are returned.

### 2.6.2. get_pdf_with_cell_length_fit_plots().
*get_pdf_with_cell_length_fit_plots()* is a function to smooth and interpolate cell length data and evaluate the resulting fits. The function creates a pdf containing plots of fits (and first derivatives) in the working directory, together with the input cell length data (*fit_plots_using_bandwidth_multiplier_X.pdf*).

The *get_pdf_with_cell_length_fit_plots()* function requires the cell length data (*cell_length_data* parameter), which are to be imported beforehand in R as a data.frame or tibble. Next, the user can specify the *interval_in_cm*, *bw_multiplier* and *output_bw_tibble* parameters of the function. In absence of user specified values, defaults will be used. In short, *interval_in_cm* is the interval used to calculate fitted cell lengths (in centimetres, default = 0.1), *bw_multiplier* allows the user to manipulate the calculated bandwidth of the data (default = 1; a number between 0 and 1 will result in a stricter fit that more closely follows the raw data, whereas a number larger than 1 will increase the smoothing) and *output_bw_tibble* will return the calculated bandwidths in a tibble when set to TRUE (default = FALSE). The bandwidth, manipulatable by the bw_multiplier parameter, is calculated within the function using the dpill function of the KernSmooth package and determines the strictness of the fit based on the distribution of the input data (Ruppert et al., 1995).

The created pdf-file with the plotted cell lengths and fit curves can be used to evaluate the cell length fits for each plant and to check the impact of a range of bandwidth multipliers on these fits. A good fit does not overly follow minor local variations in cell length, but closely fits the global profile (Figure 1b). When the *bw_multiplier* value is too low, for example 0.3, too much local variation is introduced in the fit, especially in the mature region, where cell length can be considered approximately constant (Figure 1a). Inversely, when the *bw_multiplier* value is too high, for example 3, oversmoothing occurs, particularly affecting fitted cell sizes in the meristem (Figure 1c).

In the created pdf file, also the calculated bandwidths for each individual plant are plotted in the final graph. If, for some plants, the function was unable to calculate the optimal bandwidth (e.g. when an insufficient number of cell length measurements was provided),

there will be missing data in the bandwidth plot, the concerned cell length fit plots will yield no fit and a warning message will be printed in the console of RStudio. In this case, when extracting all the fitted cell lengths in the next step, an alternative bandwidth should be provided in the *get_all_fitted_cell_lengths()* function (see next section).

### 2.6.3. get_all_fitted_cell_lengths().
The function *get_all_fitted_cell_lengths()* returns the fitted cell lengths throughout the growth zone, using the same method as the *get_pdf_with_cell_length_fit_plots()* function. It has some of the parameters with the same default and meaning as in the *get_pdf_with_cell_length_fit_plots()* function, that is *cell_length_data*, *interval_in_cm* and *bw_multiplier*. Additionally, it has the *alternative_bw* and *tidy_cell_lengths* parameter.

*alternative_bw* allows the user to set an alternative bandwidth which is used for plants for which no bandwidth could be calculated (default = 0.5). Users can determine this alternative bandwidth by using the output of the *get_pdf_with_cell_length_fit_plots()* (*output_bw_tibble* as TRUE), which will cause the function to return all calculated bandwidths. The mean of the returned bandwidths usually is a suitable alternative bandwidth.

Next, the *tidy_cell_lengths* parameter controls the output of the *get_all_fitted_cell_lengths()* function and is TRUE by default. This setting causes the function to return the fitted cell lengths in a tidy format, which is the format that is required as input for the *kinematic_analysis()* function. Setting *tidy_cell_lengths* to FALSE will return the cell lengths in a wide, more human readable, format.

### 2.6.4. kinematic_analysis.
When mean LERs and fitted cell lengths for each plant are obtained, the kinematic analysis can be performed using the *kinematic_analysis()* function. The function requires the LER means output of the *calculate_LER()* function and tidy cell lengths output of the *get_all_fitted_cell_lengths()* function as input (as tidy tibbles), together with meristem sizes (*meristem_size_micrometre* parameter) as a data.frame or tibble. The meristem sizes should be imported into R beforehand. Hereafter, the function performs all the kinematic calculations for each plant present in the tidy cell lengths tibble. It is therefore necessary that these plants are also represented in the LER and meristem size data, where they need to have exactly the same plant IDs. For each plant ID, the function collects the LER, cell lengths and meristem size. Hereafter, it performs all calculations

involved in a kinematic analysis, previously described in detail (Nelissen et al., 2013; Rymen et al., 2010; Sprangers et al., 2016). These calculations were implemented as functions (formulae 2–13 in the user manual, Supplementary File S1) defined in the *functions_needed_by_kinematic_analysis.R* script inside the package.

### 2.7. Situational errors

In order to address errors or difficulties users are experiencing, inherent to the use of R and data files, we maintain an overview of user specific errors/difficulties and how to cope with them in the README.md file of the *leafkin* repository on the IMPRES-lab GitHub page.

## 3. Results

With the aim of making kinematics data analysis more accessible, we illustrate the use of the *leafkin* package on a recently published data set that was obtained in an experiment where maize seedlings were exposed to a control, a mild (46.5 mg Cd kg$^{-1}$ dry soil) and a severe (372.1 mg Cd kg$^{-1}$ dry soil) cadmium treatment, resulting in an inhibition of leaf elongation rate by 24 and 46%, respectively (Bertels et al., in press). The data are provided as a set of tab-delimited text files on the IMPRES-lab GitHub page (https://github.com/impres-lab). Plant IDs include reference to the treatments: control (C), mild (M) and severe (S), respectively. The treatment identifier is followed by the plant number. Together with these data, a tutorial script is provided on the IMPRES-lab GitHub, which, in conjunction with the user manual (Supplementary File S1), will quickly familiarize the user with the dataset structure and the possibilities of the *leafkin* package.

The analysis of kinematics data first involves the processing of leaf length measurements to obtain leaf elongation rates. Then, cell length data, obtained from the microscopy study, are analysed and processed in order to obtain the smoothed and interpolated cell length profile for each plant. Finally, the leaf elongation rates, estimated cell length profiles and meristem sizes are used to perform the kinematic analysis for individual plants.

### 3.1. Calculating average LERs

Leaf elongation rates are calculated using *calculate_LER()*. In maize, we typically dissect leaves 3 days after they have emerged from the whorl of older leaves, yielding three daily leaf length measurements. The remaining plants were tracked until they reached their final leaf length and have more measurements. For the dissected plants, the first three leaf length measurements can be used to calculate two LERs. For this reason we set *n_LER_for_mean* equal to 2 (default value) and *output* to 'means' (default value), which causes the *calculate_LER()* function to use only the first two LERs to calculate the mean LER of each plant and return it (Table 2a).

### 3.2. Evaluating the fitting of cell length profiles

Individual cell length measurements and their position (see file description in methods) are used to determine the fitted cell length at every interval location along the leaf axis. Before extracting the fitted cell lengths (in the next step), the quality of the fit should be evaluated using the *get_pdf_with_cell_length_fit_plots()* function. This function creates a pdf file containing plots of all fitted cell lengths and their first derivatives in the working directory. Inspect-

ing these plots allows assessment of the quality of the fit. For the interval parameter, we have set *interval_in_cm* to 0.1 cm, which resulted in cell lengths estimated at every millimetre. The default bandwidth multiplier of 1 (*bw_multiplier* parameter) resulted in a good fit (Figure 1b). Finally, the absence of a warning message and presence of a fitted cell length profile in all plots indicate that all bandwidths were successfully calculated.

### 3.3. Fitting cell length profiles

After checking the fitted cell lengths profiles, we retrieve the fitted cell lengths using the *get_all_fitted_cell_lengths()* function. For this, we use the same cell length measurements and parameter settings as in the *get_pdf_with_cell_length_fit_plots() function*. If needed, the mean of the calculated bandwidths can be used in the *alternative_bw* parameter if some bandwidth calculations failed. After running the function, the resulting fitted cell lengths are stored (as a tidy tibble) for use in the *kinematic_analysis()* function. Besides the use of these cell lengths in the *kinematic_analysis()* function, this data can also be used to calculate and plot average cell length profiles with error bars (Figure 2).

### 3.4. Kinematic analysis

Using the mean LERs and fitted cell length profiles for each plant obtained in the previous steps, combined with the measured meristem sizes, we next perform the actual kinematic analysis using the *kinematic_analysis()* function. Using this function, we perform all kinematic calculations simultaneously and obtain the results for the following parameters in a tibble: leaf elongation rate (LER, mm hr$^{-1}$), length of the meristem (mm), length of the elongation zone (mm), length of the growth zone (mm), length cells leaving meristem ($\mu$m), mature cell length ($\mu$m), number of cells in meristem, number of cells in elongation zone, number of cells in total growth zone, cell production rate (cells hr$^{-1}$), cell division rate (cells cell$^{-1}$ h$^{-1}$), relative cell elongation rate ($\mu$m $\mu$m$^{-1}$ hr$^{-1}$), cell cycle duration (hr), time cells spend in the meristem (hr), time cells spend in the elongation zone (hr) (Table 3a). Note that the LERs presented in this tibble are only the LERs of the plants involved in the microscopy study (i.e. the plants on which the kinematic analysis was performed). With the kinematic analysis completed, the results can be presented in a table, summarising the values as means plus standard error, whilst comparing treatments, genotypes, and so on as percentages compared to the reference treatment (Table 3b).

## 4. Discussion

Kinematic analysis allows to relate spatial-temporal variations in rates of cell division and/or expansion to growth of different types of plant organs. These analyses have been adapted to the growth pattern of specific organs, but generally their application involves laborious, manual image analysis and data processing. This has presumably hampered their wider use. A number of tools have been developed to automate the image analysis of time-lapse images of growing root tips, allowing the analysis of cell expansion profiles (van der Weele et al., 2003; Walter et al., 2002) and the extraction of cell size distributions along an axis (Pound et al., 2012) or in three-dimensional structures (Barbier de Reuille et al., 2015; Pound et al., 2012), based on which dynamics of cell division and expansion can be determined. Although kinematic analyses of cell division and expansion along the axis of root tips (Erickson &

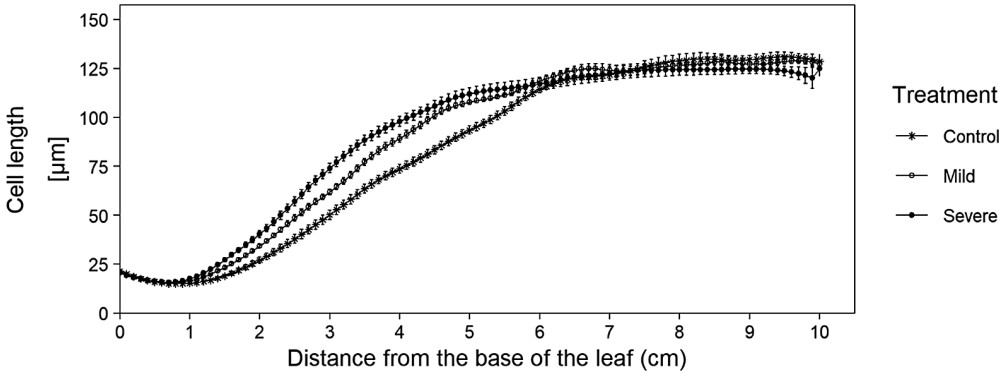

**Fig. 2** Averages of fitted cell length profiles for three cadmium treatments. Data originate from the sample dataset in which we analysed leaf growth of B73 plants, grown in control and cadmium spiked potting soil (mild and severe treatment), (Bertels et al., in press). This graph illustrates the added advantage of being able to plot cell length curves, since this plot illustrates that mature cell length is not affected by our treatment, however the growth zone size is affected (i.e. under cadmium conditions, cells reach their mature cell length closer to the base of the leaf). The code to recreate this plot is available in the tutorial R-script (https://github.com/impres-lab).

**Table 3.** Kinematic analysis output.

| A. | | | | | |
|---|---|---|---|---|---|
| plant_id | LER | meristem_size | length_of_growth_zone | length_of_cells_leaving_meristem | relative_cell_elongation_rate |
| C.1 | 3.145833 | 1,2423 | 72 | 17.00895 | 0.045889 |
| C.10 | 3.1875 | 1,5500 | 82 | 19.02419 | 0.044031 |
| C.2 | 3.208333 | 1,4792 | 66 | 17.56982 | 0.050553 |
| C.4 | 3.291667 | 1,2350 | 62 | 18.24741 | 0.054594 |
| C.7 | 3.125 | 1,4568 | 71 | 18.79564 | 0.049643 |

| B. | | | | |
|---|---|---|---|---|
| Parameter | Control | Mild | Severe | Percentage change in mild/severe stress |
| Final leaf length (mm) | 761 ± 16 | 634 ± 26 | 576 ± 47 | -17* / -24* |
| Leaf elongation rate (mm·h⁻¹) | 3.23 ± 0.03 | 2.47 ± 0.05 | 1.74 ± 0.07 | -24* / -46* |
| Length of the meristem (mm) | 14.3 ± 0.7 | 12.2 ± 0.5 | 10.6 ± 0.5 | -15* / -26* |
| Length of the elongation zone (mm) | 56 ± 3 | 51 ± 3 | 48 ± 4 | -8 / -14 |
| Length of the growth zone (mm) | 70 ± 3 | 64 ± 3 | 59 ± 4 | -10 / -16 |
| Length cells leaving meristem (μm) | 18.0 ± 0.4 | 18.7 ± 0.4 | 18.5 ± 0.6 | +4 / +3 |
| Mature cell length (μm) | 129 ± 3 | 127 ± 2 | 123 ± 3 | -2 / -4 |
| Number of cells in meristem | 873 ± 43 | 720 ± 36 | 618 ± 32 | -17* / -29* |
| Number of cells in elongation zone | 999 ± 22 | 881 ± 31 | 829 ± 47 | -12 / -17* |
| Number of cells in total growth zone | 1872 ± 52 | 1602 ± 24 | 1448 ± 46 | -14* / -23* |
| Cell production rate (cells·h⁻¹) | 25.0 ± 0.7 | 19.6 ± 0.4 | 14.2 ± 0.2 | -21* / -43* |
| Cell division rate (cells·cell⁻¹·h⁻¹) | 0.029 ± 0.002 | 0.028 ± 0.001 | 0.023 ± 0.002 | -5 / -19 |
| Relative cell expansion rate (μm·μm⁻¹·h⁻¹) | 0.049 ± 0.002 | 0.043 ± 0.002 | 0.033 ± 0.002 | -13 / -33* |
| Cell cycle duration (h) | 24 ± 1 | 26 ± 1 | 30 ± 2 | +5 / +25* |
| Time cells spend in the meristem (h) | 238 ± 15 | 242 ± 13 | 282 ± 20 | +2 / +19 |
| Time cells spend in the elongation zone (h) | 40 ± 2 | 45 ± 2 | 58 ± 3 | +12 / +45* |

**A.** Kinematic analysis data in R from individual plants after running the *kinematic_analysis()* function. **B.** Statistically processed kinematics data, as an illustration on how the final data set after analysis in R can be presented (Bertels et al., in press). Data shown are mean values plus standard error, where the percentage in the right column indicates differences relative to the control treatment, where an ∗ indicates a significant difference ($p < 0.05$).

Sax, 1956; Goodwin & Stepka, 1945) and monocotyledonous leaves (Hans Schnyder & Nelson, 1987; Volenec & Nelson, 1981) have been performed for decades, to our knowledge, no tools have been developed to automate the kinematic analysis of this type of organ.

To address this, the *leafkin* package provides a user-friendly automation of the workflow of the kinematic analysis of mono-

cotyledonous leaf growth and makes this analysis more accessible and reproducible than before. In combination with recent publications describing in detail the practical methodology (Nelissen et al., 2013; Rymen et al., 2010; Sprangers et al., 2016), this package provides an additional tool to facilitate this analysis. It provides several benefits:

The analysis of LER not only provides the basis for the kinematic analysis of cell division and expansion, but can also be used independently to analyse longitudinal growth dynamics of monocotyledonous leaves (and other linear growing organs such as coleoptiles, hypocotyls, stem internodes, root tips) based on length data in function of time. The use of the *calculate_LER()* function omits the tedious task of calculating all time intervals and corresponding leaf elongation rates for each plant and allows for easy processing afterwards in R. The automated calculation of leaf elongation rates and time intervals is particularly useful when, for some plants, data are missing, and growth and time intervals have to be adjusted accordingly (Table 2d).

Concerning cell length profiles, the user immediately obtains an overview of all cell length plots in one file for easy screening of the quality of the fit (using the *get_pdf_with_cell_length_fit_plots()* function), where in the past, these plots would have been created individually. Also, all the fitted cell lengths are created at once and can immediately be stored in one tibble after running the *get_all_fitted_cell_lengths()* function, which allows the user to easily create cell length plots and visualize differences in meristem size, growth zone size, cell elongation and mature cell length (Figure 2).

Once mean LERs and cell length fits are obtained, the next step is to combine the calculated LERs with the fitted cell length profiles and meristem sizes and perform the same set of calculations for each plant in the experiment. Using the *kinematic_analysis()* function, all these calculations are automatically performed for all plants at once, where a manual analysis would have taken significantly longer [e.g. for the example experiment, manual analysis would take an entire day, where calculation through the *leafkin* package would be finished under an hour (incl. data file preparation and quality control)]. Next to time saving, manual data processing creates room for human error, where the use of the package prevents this.

Finally, and most importantly, using the package does not require in-depth knowledge of the underlying mathematics, making kinematics available for a broader audience of molecular and developmental biologists.

### 4.1. Critical remarks

The *calculate_LER()* automatically calculates mean LERs using a given number of calculated LERs, starting from the first LERs available for each plant. We have opted for this because, in our experience with rice and maize, the growing leaf is in its steady-state growth (i.e. when leaf elongation rates and cell length profiles are approximately stable for a significant period) when it emerges from the whorl of older leaves and it maintains this quasi constant growth rate for several days. However, it is worthwhile to inspect the LER curve of the leaf over time to verify that its growth is approximately steady at the start of the leaf length measurements for other species and treatments. Setting the *output* parameter to *tidy_LER* and plotting the LERs over time will allow this. When, in the species of interest, steady-state growth occurs later, the individually calculated LERs can be used to calculate the appropriate steady-state mean LERs outside the package (set the *output* parameter of *calculate_LER()* function to *tidy_LER*). These means can then be provided as a tidy tibble by the user to the *kinematic_analysis()* function. If no steady-state is observed (e.g. the LER progressively decreases after emergence), then the LER calculated over the first time-interval is the best approximation. Incorporating non-steady-state behaviour requires additional time points for the cellular

analysis to include time-dependent changes in the cell length profile in the kinematic equations (Beemster & Baskin, 1998; Silk, 1992). This is currently not supported by the *leafkin* package. For non-steady growing situations, including coleoptiles, hypocotyls and stem internodes, *calculate_LER()* and *get_all_fitted_cell_lengths()* are still useful for whole organ growth analysis and obtaining cell length profiles respectively. However, violation of the steady-state assumption and in case of coleoptiles and hypocotyls, the absence of a cell division zone, *kinematic_analysis()* is not suitable for the calculation of cellular parameters for these organs.

Automated data analysis cannot overcome mistakes in data collection and entry. The functions do check the input for structure and data format, but not whether the provided values make sense. It is therefore the responsibility of the user to monitor the quality of the data used. When, for instance, the cell length plots for a particular leaf do not look fluent, it is worth comparing the cell length profile and obtained kinematic results with other leaves in the same experiment, to evaluate their reliability. Also, the cause for outliers in the results of one or more parameters for a specific plant can potentially be revealed by evaluating the input data and cell length fits.

Finally, as a general note on the kinematic analysis of monocot leaves: the kinematic analysis described here is based on epidermal cell length measurements and meristem sizes determined by observing DAPI stained nuclei in the epidermis. The kinematic analysis therefore intrinsically represents the organ as a longitudinal file of cells of a well-defined cell type, in our case epidermal pavement cells located adjacent to stomatal files. Molecular, metabolite and other analyses of the corresponding zones can provide important insight into the underlying regulation of cell division and expansion (summarized by Sprangers et al., 2016). However, when whole tissue analyses (e.g. flow cytometry and quantitative polymerase chain reaction) are used and related to meristematic activity, small discrepancies can be observed when compared to the meristem size estimation in the kinematic analyses (based on DAPI stained nuclei of epidermal cells only). Specifically, epidermal cells in the monocot leaf growth zone are known to transition to cell elongation while underlying cell types are still undergoing cell division (Bertels et al., in press; Huybrechts et al., 2020; Tardieu et al., 2000). This should be taken into account when zones sizes, obtained through a kinematic analysis, are used to situate results of whole tissue analyses.

### 5. Conclusion

Kinematic analysis already exists for nearly a century (Goodwin & Stepka, 1945). The technique has been used extensively to investigate cell division and expansion in root tips (Gázquez & Beemster, 2017) and the technique has progressively been finetuned (Nelissen et al., 2013; Rymen et al., 2010; Sprangers et al., 2016). With the *leafkin* package, we provide a tool for the automation of kinematic data analysis for monocotyledonous leaves. Raw data can be processed significantly faster and with less room for human error. Moreover, separate parts of the package can be of use. For example, the *calculate_LER()* function can be used to automatically calculate LERs for large sets of plants. Through providing a limited set of functions, in addition to the already extensively described methodology, we believe that *leafkin* makes kinematic analysis of monocotyledonous leaves more accessible than before, which can result in a more widespread and frequent application of this technique to rigorously quantify the cellular basis of leaf growth.

## Acknowledgements

We would like to thank the members of the IMPRES research group and master students of the biology master programme (Antwerp University) for testing the library and providing feedback.

**Financial Support.** This work was supported by the Research Foundation Flanders (FWO) by project funding for J.B. [G0B6716N].

**Conflicts of Interest.** The authors declare no conflicts of interest.

**Authorship Contributions.** J.B. wrote the manuscript, created the tables and figures and created the *leafkin* package and tutorial script. G.B. supervised the work and edited the manuscript.

**Data and Coding Availability Statement.** The *leafkin* package, sample data and tutorial script are available on the IMPRES lab GitHub page, https://github.com/impres-lab.

**Supplementary Materials.** To view supplementary material for this article, please visit http://dx.doi.org/10.1017/qpb.2020.3.

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
