## [Reviewer Report]

Dear Editor,

We are very pleased to submit our manuscript "leafkin – An R package for automated kinematic data analysis of monocot leaves" for evaluation to be published in the journal of Quantitative Plant Biology. Kinematic analysis is a rigorous numerical approach to quantify the contribution of cell division and expansion parameters in organ growth. A relatively small community has developed and applied this extremely powerful analysis for nearly a century now. Efforts from our own and other laboratories over the years to promote it's use, by providing extensive reviews of the mathematic basis and the experimental procedures, has been pushing a slow increase of its use. We think that the absence of a user-friendly tool to perform the complex data analysis involved, is currently the biggest hurdle for it's wider usage. Therefore we developed an R-package, we called leafkin to do just that. In this manuscript we describe it's basis, it's use and the results obtained from the analysis of an example dataset that is provided online to allow future users to work with the package prior to processing their own dataset. The whole package, including example data have been made available through GitHub, which has the advantage that we can update it in future. 

We think this topic fits perfectly in the scope of this new journal and this was confirmed in discussion with the editor-in-chief who commissioned it. Therefore we hope that it will be positively evaluated and we're looking forward to your feedback.

---

## [Reviewer Report]

*Comments to Author*: Bertel and Beemster present an R package to perform kinematic analysis of the leaves of monocots. The package has 4 functions that can be used for the automatic estimation of cell division and expansion rates, and ultimately for the kinematic analysis. The advantages of the “leafkin” R package include a significant reduction in the time spent by the user in the data analysis, as well as in the reduction of human-introduced errors during this otherwise laborious process.

I would like the authors to consider the following issues:

- The authors need to introduce in more detail the kinematic analysis. This is important to highlight how this method provides quantitative information of the dynamics of plant growth, particularly for non-familiar readers to fully appreciate the advantages of the “leafkin” R-package.

- It is also important to provide the formulas that are implemented by the R functions in the Methods section. Although the authors mention that this has been revised in previous publications, it is important to present this in the current manuscript to understand the calculations that are being performed by “leafkin”.

- It will be helpful to include a figure of the general pipeline of “leafkin” and how the functions are connected with each other.

- The authors make no mention in the introduction nor the discussion about previously published tools for the automatic kinematic analysis of plant growth. All references of tools for kinematic analysis must be included, including a comparison of “leafkin” with these previous contributions.

- The authors mention that leafkin functions can be used individually (e.g. calculate_LER() ). Could the functions also be used to analyse growth in other plant organs where cells are arranged in linear files? If so, mentioning how the protocol can be adapted for the kinematic analysis of other organs will showcase the wide applicability of “leafkin” for the automatic analysis of plant growth.

Other comments:

- Line 56. Explain what “lowering the entry level to the package” means.

- Line 163. It will be useful to explain the intention of a “tidy” table when it is first introduced, rather than in the discussion (line 337).

- Table 2. The presentation of the tables should be improved. I understand the idea from the text, but the red arrows are confusing.

- Figure 1 caption. Instead of “A more strict” consider using “A very strict”.

- Line 255. The references to the code, e.g. “R-script section 0.4.1 + 3.A” are not very intuitive.

- Line 361. Please expand on the “small discrepancies” mentioned.

- What function is used to create the plot in Figure 2?

---

## [Reviewer Report]

*Comments to Author*: Review of the manuscript QPB-20-0006, titled „leafkin – An R package for automated kinematic data analysis of monocot leaves”, by Bertels and Beemster, submitted to Quantitative Plant Biology.

The manuscript describes an original R-software tool for kinematic analysis of a grass leaf. The tool is dedicated to assist researchers in fast and the least person-biased calculations of leaf growth parameters that would be comparable with data obtained by others. Being relatively straight-forward and accompanied by a user-friendly instructions with examples, the tool has the potential to be used by many plant biology labs. In my opinion, some minor issues need to be addressed by the authors in order to improve the manuscript. They are listed below in order of appearance.

l.76 – I would replace “the first 10 cm of a leaf” by referring to the basal leaf portion (the apical portion is the first to appear and differentiate so at present this is misleading)

l.85-99 – some of these details are repeated in the Table 1 legend, please consider removing repetitions

table 1 – in the legend you could briefly explain the difference between char, int or double types

l.137, 240 – the paper under review is referred to, but the reader will have little clue how to find it after publication. Have you perhaps stored the manuscript in some archives? It may be worth considering to give some additional information about this publication, or where the information could be found

l.155 – I am not sure if “modern take on” is a correct phrase. Could it be replaced by “a modern format”?

l.162 – what you mean by tidy and wide?

Table 2 – what are the units of parameters shown here? Maybe provide this information in the legend

l.183, 196 – it would help the reader if the term “bandwidth” were explained, and the way how the optimal bandwidth is calculated were briefly described

figure 1 – can the user obtain parameters of the fitted curves that would allow him to compare the curves from different plants?

l.224-5 – is the style OK here?

l.227-8 – because the present manuscript is not long you could briefly explain how the kinematic analysis is performed, especially since the first two citations referred to are book chapters not of open access

Figure 2 – would be more readable if colors differed between the treatments

l.282-97 – am I correct that cellular parameters (cell numbers, production rates etc.) actually apply to a longitudinal cell file extending from the leaf base to the apex ? Whether yes or no, in my opinion this needs to be clearly stated

In Discussion: Could you perhaps address the question whether leafkin could be also used to analyze kinematics of other “linearly” growing organs, like coleoptile, hypocotyl or stem internode, and what would be limitations or critical remarks for users?

Finally, I found some typos:

l.26 – a comma or & is missing between the two cited papers

l.32 – a comma after “zones” is unnecessary

l.371 – “e” is missing in “monocotyledonous leaves”

---

## [Reviewer Report]

*Comments to Author*: Dear Prof. Beemster,

My apologies for taking so long to get back to you on this manuscript. 

Both reviewers have positive comments on the work, and made a series of relatively minor suggestions to improve the manuscript. If you could please address these concerns for resubmission, we will be happy to consider the work for publication.

In addition to the flow chart illustrating the workflow of the tool, it would be helpful to include a user guide as a supplemental item to guide users as to its use. This is particularly important for uptake in terms of preserving technical aspects for things such as file paths, and troubleshooting any errors. I appreciate this is in the text, though a protocol-like text would also be useful.

Thank-you for considering Quantitative Plant Biology for the publication of your work.

Best wishes

George

---

## [Reviewer Report]

Dear editor, 

We want to sincerely thank you for the appreciation expressed towards the submitted manuscript and for your invested time and effort to provide valuable feedback to refine its contents. We carefully considered all points raised by you and the reviewers and accordingly made adjustments, clarifications and additions to the manuscript. 

We have listed all points and added our response to each of them. Where appropriate, we made reference to the adjustments made in the updated manuscript. Extra effort was invested in providing a protocol-like text, highlighting the technical aspects for things such as file paths, and troubleshooting any errors. Furthermore, in the protocol, a flow chart illustrating the workflow of the tool was incorporated.

We think these changes have significantly improved our manuscript and hope it is now acceptable for publication in your exiting new journal.

Sincerely,

Jonas Bertels, Gerrit TS Beemster

---

## [Reviewer Report]

*Comments to Author*: Review of the revised manuscript QPB-20-0006, titled „leafkin – An R package for automated kinematic data analysis of monocot leaves”, by Bertels and Beemster, submitted to Quantitative Plant Biology.

As I have already reviewed the first version of this submission, below I focus only on the changes introduced by the Authors.

First of all, the Authors very carefully revised the manuscript. It is now greatly improved, although the first version was already very good. Second, the idea of adding the manual as a supplementary file, and providing there all the technical information, equations, step-by-step guide, as well as explanation of terms used in the main text, was really a very good idea, although one can imagine how much work it required to be prepared. Nevertheless, in such form, i.e. the main text plus extensive manual, the publication will in my opinion ensure the leafkin usage by plant biologists, from which both the Authors, and the “new born” journal will benefit, for sure.

---

## [Reviewer Report]

*Comments to Author*: Dear Gerrit and Jonas

Thank-you for your submission and addressing the issues raised by the reviewers and myself.

We are happy to accept this manuscript for publication, and apologise for the delays during the review process.

We will be in touch with further information shortly.

Regards

George Bassel